

# Brief communication: Sharp winter precipitation transition on the southern edge of the Tibetan Plateau

Titouan Biget[1], Fanny Brun[1], Walter Immerzeel[2], Léo Martin[3], Hamish Pritchard[4], Emily Collier[5], Yanbin Lei[6, 7], and Tandong Yao[6, 7]

[1]Université Grenoble Alpes, CNRS, INRAE, IRD, Grenoble INP, IGE, 38400 Saint-Martin-d'Hères, France
[2]Department of Physical Geography, Utrecht University, Utrecht, The Netherlands
[3]Aix Marseille Univ, CNRS, IRD, INRAE, CEREGE, Aix-en-Provence, France
[4]British Antarctic Survey, Cambridge, United Kingdom
[5]Department of Atmospheric and Cryospheric Sciences (ACINN), University of Innsbruck, Innrain 52, Innsbruck, 6020, Austria
[6]Key Laboratory of Tibetan Environment Changes and Land Surface Processes, Institute of Tibetan Plateau Research, Chinese Academy of Sciences, Beijing 100101, China
[7]CAS Center for Excellence in Tibetan Plateau Earth System Sciences, Beijing, 100101, China

**Correspondence:** Titouan Biget (titouan.bgt@gmail.com)

**Abstract.**

The Tibetan Plateau is a high-altitude arid region, where limited in-situ precipitation measurements are available. In this communication, we document a strong precipitation gradient at the southern edge of the Paiku Co catchment (southern Tibetan Plateau) from in-situ data and atmospheric model outputs. In particular, we use water pressure time series from proglacial
5    lakes, two automatic weather stations, and data from ERA5-Land reanalysis and CORDEX-FPS-CPTP ensemble. We show that precipitation can vary by one order of magnitude over a short distance of 10 km in a rather smooth terrain throughout the winter and the pre-monsoon season. This large precipitation gradients marks the transition between the great Himalayas and the Tibetan Plateau.

## 1 Introduction

The Tibetan Plateau is a high-altitude region characterized by a dry and arid climate, and by the presence of multiple lakes and glaciers (Yang et al., 2014). Most of the lakes of the Tibetan Plateau have been expanding rapidly since the mid-1990s, representing an additional terrestrial water storage of 6 to 9 Gt yr$^{-1}$ (Zhang et al., 2017). However, some lakes located on the southern edge of the plateau have shrunk for the last three decades (Lei et al., 2018; Zhang et al., 2021). The changes in lake
15   volume are mostly attributed to decadal changes in precipitation (Zhang et al., 2021). Precipitation measurements are scarce on the Tibetan Plateau, and thus analysis of climate and meteorology relies largely on reanalysis, remote sensing products and climate models (e.g., Collier et al., 2024).



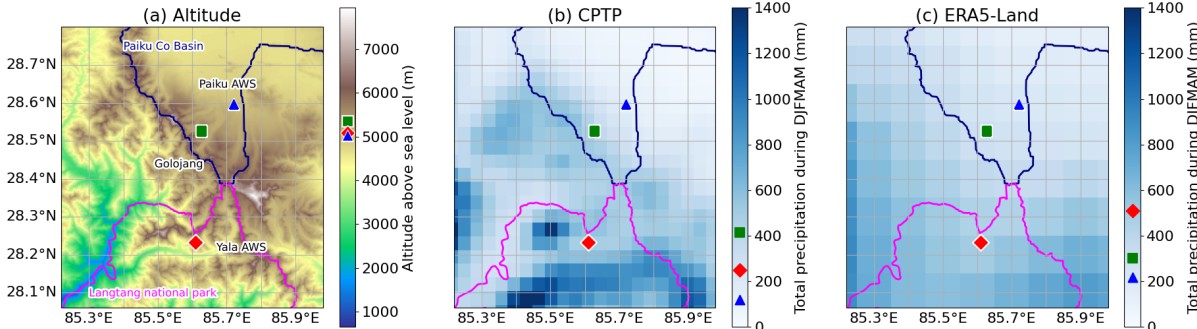

**Figure 1.** Elevation and spatial distribution of precipitation above the studied area for CPTP and ERA5-Land models from October 2019 to May 2022. The symbols represent the locations of the in-situ recordings on the maps and their interpolated values on the color bars. The southernmost and brightest line marks the boundary of the Langtang National Park. The northernmost and darkest line marks the boundary Paiku Co basin.

The challenges associated with precipitation measurements are numerous in mountainous regions, especially when a substantial fraction falls in solid forms. Precipitation gauges suffer from large underestimations of solid precipitation, 42.6% on average for unshielded gauges and 30.6% for single-Alter shielded gauges (Kochendorfer et al., 2017). These underestimations can be much larger in cases of strong winds (Goodison et al., 1998). To overcome these issues, recent studies suggested that solid precipitation could be also estimated from frozen lake pressure changes (Pritchard et al., 2021). This method relies on the assumption that the catchment is frozen, making the surface run-off and the evaporation negligible due to the cold conditions. Since lakes are used as sensors, the size of the sensor is 6 to 9 orders of magnitude above the size of common precipitation gauges, thus providing a spatially averaged estimate of precipitation that is expected to be less sensitive to undercatch and to spatial variability of precipitation (Pritchard et al., 2021), at the cost of application over only a few months of the year.

In this study, we report new observations of lake level changes during the cold season (December to May) of a catchment located on the southern edge of the Tibetan Plateau. From the lake water pressure time series, we reconstruct precipitation estimates based on Pritchard et al. (2021)'s method and compare them to conventional gauge measurements located on both sides of the orographic barrier of the Himalaya. We then compare these observations to precipitation estimates from two datasets, ERA5-Land reanalysis (Copernicus Climate Change Service) and simulations from the Coordinated Regional Climate Downscaling Experiment Flagship Pilot Study (CORDEX-FPS) Convection-Permitting Third Pole (CPTP) project (Collier et al., 2024; Prein et al., 2023), to discuss the added value of kilometer-scale convection-permitting atmospheric models.

## 2 Study site

The in-situ measurements were collected in the southern part of the endorheic Paiku basin (Fig. 1 - a), on the southern Tibetan Plateau (China). The catchment drains into Paiku Co, an alpine lake situated at 4590 m a.s.l. that has an area of approximately 270 km$^2$, which is slightly more than one tenth of the whole catchment (2376 km$^2$). Paiku Co is partially fed by the 41 glaciers





present in the basin (Lei et al., 2018). Although Paiku Co lake has shrunk for the last decades, proglacial lakes located higher up in the catchment have been growing for the past 50 years due to glacier retreat (Lei et al., 2018).

We focus on the proglacial lake named Golojang Co, which is located in the south of the catchment at an elevation of 5357 m a.s.l and has an area of 5.54 km$^2$(see Fig. 1 - a). It is frozen a large part of the year, during winter (DJF), pre-monsoon (MAM) and part of monsoon (starting in June), to finally break-up around July. The outlet of the lake flows through a moraine, then channels into the Nijile river. The lake occupies the over-deepening carved by the glacier retreat. The glaciers are still in contact with the lake, with a calving front releasing small icebergs. The surrounding orography may favor the accumulation of

snow on its local lowest areas, i.e, on the frozen surface of the lake.

    In Paiku Co catchment, for elevations below 5000 m a.s.l, the measured annual precipitation varies between 150 and 300 mm, with winter precipitation accounting for approximately 10% of the annual total (Lei et al., 2018, 2021; Martin et al., 2023). These estimates are based on an automatic weather station (AWS) located close to the main lake Paiku Co (at 4600 m a.s.l.; Lei et al., 2018), and an other AWS located 10 km south of the lake (named Paiku AWS in this study and located at 5030

m a.s.l.; Martin et al., 2023).

## 3   Data and method

To estimate the precipitation around Paiku Co catchment, we use a collection of datasets from in-situ observations and from models.

### 3.1   Precipitation from meteorological stations

We use meteorological data from two AWS located on different sides of the Himalayas: on the northern side, within Paiku Co catchment in China, we use the Paiku AWS (5033 m a.s.l., Martin et al., 2023), and on the southern side, we use Yala AWS (5090 m a.s.l., Immerzeel et al., 2014) located in Langtang National Park in Nepal (Fig. 1 - a), near Yala Peak, base camp.

    Both AWS record standard meteorological variables (air temperature and relative humidity, wind speed, incoming and outgoing radiation, and atmospheric pressure). Additionally, they measure snow thickness with a Campbell SR50 ultrasonic ranging

sensor and precipitation with an OTT Pluvio2 all-weather precipitation gauge. Paiku Co AWS started recording from 29 October 2019, and the record is uninterrupted until May 2022. Antifreeze liquid and engine oil were added to the OTT Pluvio2 bucket in order to prevent the water inside it from freezing or evaporating. Yala AWS has been operating since October 2015 but was not recording from 18 November 2019 to 27 March 2020 due to data overwriting. The record is then uninterrupted until May 2022. Antifreeze was added to the OTT Pluvio2 bucket, but no oil was added, and thus evaporation is expected.

As a consequence, the signal from the precipitation gauge was rectified by replacing the decreasing height values with neutral values (not decreasing nor increasing). Both OTT Pluvio2 are equipped with a windshield.



## 3.2 Precipitation from lake water pressure

Three Hobo U20 pressure transducers (PT) were immersed in Golojang Co and were recording from 01 November 2019 to 14 April 2022. A fourth PT placed outside the lake is used as an atmospheric pressure reference in order to correct for diurnal
changes in atmospheric pressure. The immersed PT record water pressure and temperature of the lake every half an hour. The PT were immersed at water depths ranging from 50 to 80 cm and hence could be frozen during the coldest months. Only one PT was never frozen, and consequently we analyze results from this single sensor. Because of the strong noise on the signal from the PT, we filter the series with a 48-hour rolling mean.

It has been demonstrated that such pressure time series can be interpreted as direct measurements of the precipitation falling
onto the lake during winter-like conditions (Pritchard et al., 2021). This method relies on the assumption that the surface and sub-surface runoffs are at their minimum due to freezing conditions, and limited evaporation from the lake due to its ice cover, although it does not have to be necessarily entirely frozen if the evaporation is neglected. In these conditions, pressure variations in the lake are attributed only to the precipitation over the surface of the lake and its drainage. The typical pressure signal expected is a constant slow decrease due to the drainage of the lake, with some quick jumps due to precipitation. We used
two criteria to verify the frozen assumption: i- we extrapolate the temperature from Paiku AWS to the elevation of Golojang Co, assuming an -6.5 K/km environmental lapse rate, and keep days with a mean temperature below 0°C; and ii- we map the ice cover extent of Golojang Co from Landsat 7, 8, and Sentinel-2 images using the Google Earth Engine Digitisation Tool (GEEDiT; Lea, 2018) and keep periods with more than 90% of the surface frozen.

We followed the method developed by Pritchard et al. (2021) to estimate the precipitation amount over Golojang Co. When
the lake is frozen, the pressure measured by the PT sensor ($p$ in Pa) follows the hydrostatic equation:

$$p = \rho_w g (h_w + h_{swe}) + p_0 \tag{1}$$

with $\rho_w$ (in kg m$^{-3}$) being the density of liquid water, $g$ (in m s$^{-2}$) being the acceleration of gravity, $h_w$ (in m) the depth of the sensor relative to surface, $h_{swe}$ (in m) being the water equivalent height of the snow and ice above the surface, and $p_0$ (in Pa) being an arbitrary constant. We assume that, at the time scale of a snowfall event (from a few hours to four days), changes
in $p$ (named $dp$) are solely due to changes in $h_{swe}$ (named $dh_{swe}$), due to precipitation and to changes in $h_w$ due to the lake drainage (named $dh_w$). Following Pritchard et al. (2021), we assume a constant drainage that is estimated by extrapolating $p$ just before and after the event using a linear regression on the time series fitted with a least square method. The date of the beginning, the date of the ending, and the duration of the extrapolation were determined manually. For the rest of the study, we express changes in the total water column ($dh$ in m) as:

$$dh = \frac{dp}{\rho_w g} = dh_w + dh_{swe} \tag{2}$$

We assume that the uncertainty $\varepsilon_{swe}$ on the measurement of precipitation during an event is the sum of $\varepsilon_{D1}$ and $\varepsilon_{D2}$, which are respectively the uncertainties on the regression on the drainage before and after the snowfall. The uncertainty arising from the drifting of the PT is neglected since we estimated that the drifting is small enough to be considered as a systematic bias on the scale of a single event.





## 3.3 Precipitation from atmospheric models

We use daily precipitation from the CORDEX-FPS CPTP project (Collier et al., 2024) and from ERA5-Land (Muñoz Sabater et al., 2021). The former dataset consists of simulations from 13 different models or model configurations driven by ERA5 for the hydrological year of October 2019 to September 2020 using grid spacings ranging from 2.2 to 4 km. We used model output reprojected on homogeneous grids (approximately 4 km). We use the averages on the total daily precipitation from the 13 ensemble members. ERA5-Land variables are provided on an approximately 10 km grid. We use the variable *total_precipitation* for the period October 2019 to May 2022. For both products, we extracted time series of daily precipitation for the grid point closest to the location of the lake outlet or the AWS (Figs. 1 and 3).

Our observational record is very short (<2.5 years). In order, to discuss whether the observed patterns are persistent on a climatological scale (30 years), we also investigated ERA5-Land reanalysis data from 1993 to 2022 of both monthly precipitation and trajectory of moisture parcels over the region of Nepal and Tibet.

## 4 Results and discussion

### 4.1 Hydrological cycle of Golojang Co

Golojang Co water level has a strong seasonal cycle with an amplitude of approximately 70-80 cm (Fig. 2 - a). The lake level appears to plateau at low values from late winter (February) to pre-monsoon (April), several months after freeze-up in December, as it approaches the local hydrological base level (the minimum height at the lake outlet). The water temperature increases from 1 - 2°C to a maximum of 6°C in September. The increase in lake level and temperature is linked to the larger water input when the catchment unfreezes and when the monsoon brings most of the annual precipitation (Lei et al., 2018).

Golojang Co seasonal level cycle is more pronounced than the cycle of Paiku Co (Lei et al., 2021). Paiku Co level rises sharply at the end of pre-monsoon, but its level decreases slower than Golojang Co during post-monsoon and winter (Lei et al., 2021). It is difficult to compare these lakes, because Golojang Co has an outlet, whereas Paiku Co is the sink of the basin (endorheic lake with no outlet). Evaporation is thus the only loss term for Paiku Co whereas Golojang Co experiences drainage.

### 4.2 Precipitation estimates from Golojang Co pressure time series

Using the Golojang Co pressure time series, we are able to observe 3 winters' snowfall at this previously unobserved site that lies between Yala and Paiku. We assess the conditions to apply Pritchard et al. (2021)'s method, which requires that precipitation falling in the lake catchment surrounding the lake does not run off into the lake but is stored as snow in freezing conditions. As a proxy indicator of a frozen catchment, we monitored the extent of ice cover on the lake surface and estimate the air temperature around the lake. The ice coverage percentage is known for 375 days, distributed between 20 September 2019 and 10 May 2022 (one image each 2.57 days in average). Golojang Co was at least 90% frozen during 250 of these days (Fig. 2 - a). During the hydrological years 2019 - 2020 and 2020 - 2021, the lake is partially frozen from the beginning of



December to the beginning of July, lasting for 7 months each year. The catchment is in a frozen state (i.e., the lake is at least 90% frozen and air temperature is below zero at the lake elevation) from late December to mid-June. The precipitation from lake pressure could thus be estimated for winter and pre-monsoon (DJFMAM).

The total precipitation have been estimated to $420 \pm 46$ mm for the period 01 December 2019 to 31 May 2020, $307 \pm 27$
mm for the period 01 December 2020 to 31 May 2021, and $211 \pm 9$ mm for the period 01 December 2021 to 14 April 2022 (Fig. 3 - a, b and c). Two major events that happened on 21 May 2021 and 22 March 2022 were excluded from the cumulative DJFMAM snowfalls because we suspect possible presence of surface runoff around the lake due to the high temperatures recorded at these times in the catchment. These values are probably underestimated since the events identified on the pressure time series are mostly wet spells lasting on average 73 hours. One could hypothesize that the lack of short-duration snowfalls
recorded at Golojang Co is due to the strong noise on the signal and on the rolling average applied on it that could mask small pressure variations (see discussion below).

Since no detailed study has been performed on Golojang Co before, there is only little or no knowledge about it, in particular none about its bathymetry. There is also no data about the interface between the lakes and the glaciers with which it is in contact. There is no quantitative data allowing us to estimate the impact of the calving on the pressure of the lake, but we note that
calving events are abrupt, adding mass almost instantaneously to a proglacial lake. This contrasts with the slower and more sustained signal of water-pressure change from snowfall events that typically last for hours to days (Pritchard et al., 2021). Furthermore, visual inspection of satellite images show very limited calving events on the frozen lake during the period of study.

### 4.3 Measured precipitation at the automatic weather stations

When the time series overlap, we are able to compare the Golojang winter snowfall totals to those from the Yala and Paiku AWS. During DJFMAM 2019 - 2020 (Fig. 3 a, b and c), Paiku AWS recorded only 35 mm (9% of the total precipitation at Golojang Co for the same period). The Yala base camp AWS started recording during pre-monsoon 2020, therefore 2020 - 2021 is the only meteorological year with complete data. The Yala AWS recorded a total precipitation of 289 mm during winter and pre-monsoon, while Paiku AWS only recorded 25 mm. This corresponds to 94% and 8% of the total snowfalls at
Golojang Co during the same period respectively. 226 mm of precipitation fell between 01 December 2021 and 14 April 2022 at Yala, but there was only 12 mm of precipitation at Paiku AWS, which corresponds respectively to 107% and 6% of the total snowfalls at Golojang for the same period.

While the Yala and Golojang Co snowfall observations are similar, the much lower precipitation totals recorded at Paiku AWS have the same order of magnitude as the precipitation measured close to Paiku Co (Lei et al., 2018), located 25 km to
the north, meaning that the winter and pre-monsoon precipitation in the southern and highest part of Paiku Co catchment is about ten times larger than in the lowest part of the catchment. Moreover, the transition is very sharp, because Paiku AWS and Golojang Co are located only 10 km away, and Golojang Co is only 350 m higher than Paiku AWS. A large part of the major snowfall events seems to happen simultaneously at both Yala and Golojang with a similar intensity (Fig. 3 a, b and c)). Most of the events happen at Paiku AWS too, but with much reduced intensity.



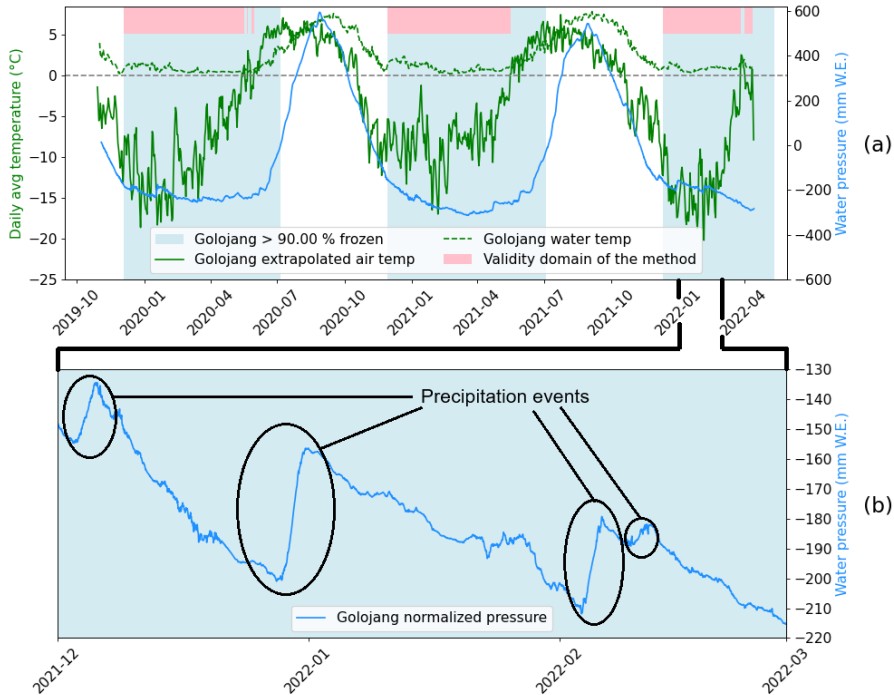

**Figure 2.** Water and extrapolated air temperature, validity domains of Pritchard et al. (2021)'s method and normalized pressure of Golojang from 31 November 2019 to 14 April 2022. Some examples of precipitation events on the pressure time series are highlighted on the bottom figure.

Both AWS were equipped with wind shields. However, both gauges probably suffer from large undercatchement of snow since they are located in windy areas. Their respective anemometers recorded that the average wind speed during snowfall in Yala is about 2.3 m s$^{-1}$ with a 0.8 quantile of 2.7 m s$^{-1}$ while the average wind speed during a snowfall in Paiku is about 5.9 m s$^{-1}$ with a 0.8 quantile of 7.2 m s$^{-1}$. In order to asses the impact of the wind on the Paiku AWS recording, we computed the ratio of Paiku AWS precipitation recorded on Golojang precipitation estimation for events with average wind speeds below quantile 0.2 (4.4 m s$^{-1}$) and above quantile 0.8, for a total of 11 events each time. In the end, we have a ratio of 28% when the wind is weak (<q 0.2) and 18% when it is strong (>q 0.8). Although it is impossible to determine precisely what is due to undercatchement and what is due to meteorological effects, it is noticeable that the most extreme winds recorded induce a variation of only 10% of the ratio, making the precipitation estimated at Golojang Co still an order of magnitude above the precipitation at the Paiku AWS.

Although the strong winds in Paiku certainly induced a large undercatchement of snowfalls, the order of magnitude of the precipitation at the AWS is in any case an order of magnitude below the precipitation at Golojang Co.



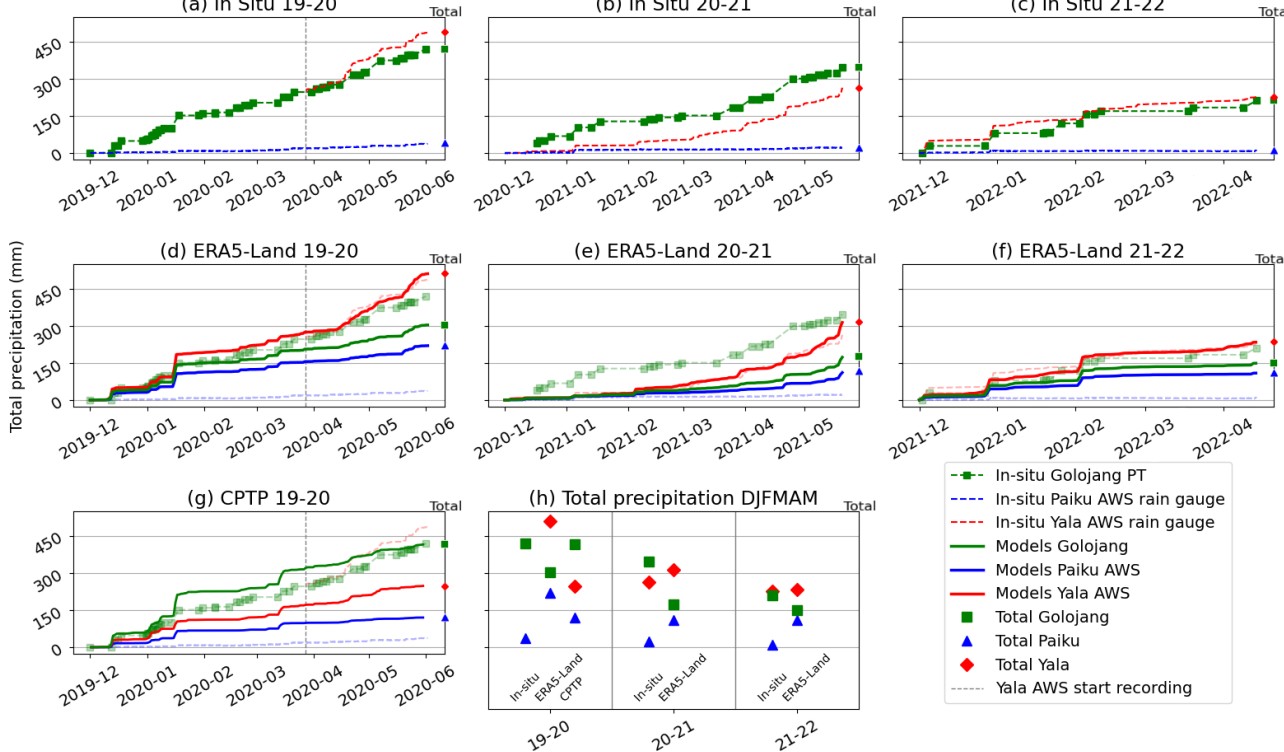

**Figure 3.** In-situ and model precipitation at the lake and both AWS locations. We note that the CORDEX FPS CPTP simulations are only available for the hydrological year of 19-20. Since Yala AWS started recording in April 2020, we choose to set its initial value to the same as in-situ Golojang during the middle of the year. The total cumulative precipitation for each period is shown on the right-hand bar of each graph and is summarized in (h).

## 4.4 Precipitation from reanalysis and atmospheric models

Both ERA5-Land and CPTP strongly overestimate the precipitation at Paiku AWS. ERA5-Land cumulative DJFMAM precipitation is, on average, 11 times larger than the in situ recording (Fig. 3 - d, e and f). CPTP is slightly closer to measurements by being only 3.4 times greater (Fig. 3 - g). The undercatch of the precipitation gauge cannot explain such a large difference. In addition, the precipitation computed by the models are far greater than the 15-30 mm winter precipitation estimated from previous studies (Lei et al., 2021, 2018).

ERA5-Land has a very good estimation of the cumulative DJFMAM snowfalls in Yala since the in situ is, on average, 1.08 times greater than the model during 2020-2021 and 2021-2022 (Fig. 3 - e and f). Unfortunately, Yala AWS was not working during the only year of the CPTP simulation. Nonetheless, since the in situ recording of the precipitation at the Yala AWS (241 mm from 27 March 2020 to 31 May 2020) has the same order of magnitude as the estimation of ERA5-Land (468 mm during the same period,Fig. 3 - d), one could notice both sources of data have nearly the same estimate of cumulative precipitation at this location during the winter and the pre-monsoon. Compared to this, the 76 mm estimation of CPTP would





be a significant underestimation compared to ERA5-Land (Fig. 3g). Finally, CPTP provides a DJFMAM cumulative snowfall

estimate at Golojang Co that differs from the PT estimation by only 1.2% during 2019 - 2020, even though CPTP slightly overestimates precipitation at the beginning of the study period and underestimates precipitation at the end compared to in situ (Fig.3 - g). ERA5-Land shows much lower precipitation at Golojang Co, with PT estimates being on average 12.3 times greater for the three years (Fig. 3d, e, and f). However, Golojang in-situ estimates are likely underestimated, as the pressure time series primarily captured wet spells averaging 73 hours, while shorter snowfalls may have been masked by signal noise.

For comparison, during the period from 01 December 2020 to 31 May 2021 (the only complete meteorological year during which all sensors were working), there was 26 events recorded at Yala AWS by the AWS, of which 18 lasted less than 2 days and had an accumulation less than 10 mm. At Golojang, the pressure time series allowed us to identify 23 events in the same period, of which only 7 could be matched with Yala events.

## 4.5   Spatial distribution of precipitation

ERA5-Land and CPTP simulations show different patterns of DJFMAM precipitation over the southern Paiku Co basin and the northern Langtang national park. Due to the finer grid resolution, CPTP models can have the ability to better capture the influence of the complex terrain on precipitation patterns than ERA5-Land (Collier et al., 2024). In this study area CPTP models have a larger spatial variability than ERA5-Land. In particular, they show relatively high precipitation on the highest parts of the study area, mainly above 5000 m a.s.l., and on the southern slopes of the Himalayas. In contrast, ERA5-Land

predicts a relatively smooth precipitation gradient, with DJFMAM precipitation decreasing from southwest towards northeast (Fig. 1b and c).

Our spatially limited in situ observations, and the short duration of CPTP runs (one year), do not allow to conclude whether CPTP models or ERA5-Land reanalysis simulate a better distribution and amount of winter precipitation on the southern edge of the Tibetan plateau. Still the ability of CPTP models to simulate sharp precipitation gradients over short distances is

promising because our results suggest that such gradients can exist.

## 4.6   Climatological analysis

Our results suggest non-negligible precipitation during DJFMAM at Golojang Co, and thus could contribute more to glacier accumulation than previously extrapolated from ERA5-Land and AWS located on Paiku Co catchment (Martin et al., 2023). As our method is applicable only during winter conditions, we cannot conclude whether there is such a sharp precipitation

gradient during monsoon, or in other words such high precipitation during the wettest season. In order to investigate whether precipitation gradients are similar during monsoon, we can only rely on ERA5 and ERA5-Land reanalysis that cover a long record (at least thirty year of data).

The ERA5-Land monthly precipitation from 1993 to 2023 reveal a highly marked seasonal pattern, with precipitation at Paiku AWS being 0.5 - 0.6 times the precipitation at Yala AWS during the winter and slowly decreasing to 0.3 - 0.4 during

the monsoon, then increasing back progressively. The ratio of the precipitation between Paiku AWS and Golojang Co does not vary much, with only 20% variation between its minimum in August and its maximum in April. The pattern of integrated water



vapor flow over the entire column in ERA5 reanalysis is consistent with this observation: in winter and pre-monsoon, there is a strong, very homogeneous north/south gradient, while during the monsoon (JJA) the pattern is much more heterogeneous over Nepal. Such results could be interpreted as seasonal variations of the spatial distribution of the precipitation over the southern Tibetan Plateau and northern Nepal, with once again, a stronger precipitation gradient during winter and pre-monsoon.

## 5 Conclusions

In this communication, we studied the hydro-meteorology of the northern Langtang National Park in Nepal and the Southern Paiku Co basin in Tibet using in-situ and modeled data from conventional and new methods. Using Pritchard et al. (2021) recent method, we estimated the snowfalls over the proglacial lake Golojang from 12 December 2019 to 09 April 2022, in the southern part of Paiku Co basin during the period DJFMAM. With this method, we converted the pressure time series of the lake that were recorded into estimates of the cumulative snowfalls when the lake is frozen. However, because of the strong noise on the signals, it may be possible that some events were not detected because of their small magnitudes. Our results highlight the potential of lakes to be used as precipitation gauges, which is especially valuable in data-scarce regions.

The snowfalls over the lake were compared to the precipitation recorded by AWS located on the southern side of the Himalaya (Yala Peak AWS in Langtang National Park) and on the Paiku Co catchment. During the three years of recording, it seems that Golojang Co cumulative snowfalls during DJFMAM are closer to Yala AWS, with a value of 200 - 400 mm, than to Paiku AWS (5 - 35 mm) despite it is located much closer to the lake.

We compared our results with precipitation data computed by reanalysis and atmospheric models (ERA5-Land and CORDEX-FPS CPTP). Both products largely overestimate the precipitation inside the Paiku Co basin compared to the automatic weather stations recordings and our estimates based on pressure variations. Still, CPTP estimates are consistent with the DJFMAM precipitation recorded at Golojang during the meteorological year 2019 - 2020. ERA5-Land DJFMAM cumulative precipitation are surprisingly very close to the recording of Yala AWS.

The in-situ data we gathered are thus not matching any model at all three locations at once. However, CPTP shows the best ability to reproduce the spatial distribution of the precipitation as it is recorded between the lake and the AWS . On the other hand, ERA5-Land was not matching any of the precipitation recording in the Paiku Co basin.

An analysis of the long term precipitation was performed over the three sites using ERA5-Land data from 1993 to 2022. Even though we show that ERA5-Land is not the better fit for a study on such a scale, it revealed a seasonal variation of the spatial distribution of the precipitation, with a stronger gradient between Yala and the Paiku basin during the coldest part of the year (Dec. to May). This analysis suggests that we cannot extend our findings about precipitation gradients outside the cold season, due to different moisture origin and circulation systems.

*Code and data availability.* The code used to calculate precipitation and its uncertainties can be accessed together with the event dates and Golojang Co pressure time series at https://doi.org/10.5281/zenodo.14894770, (Biget, 2025)



## Appendix A

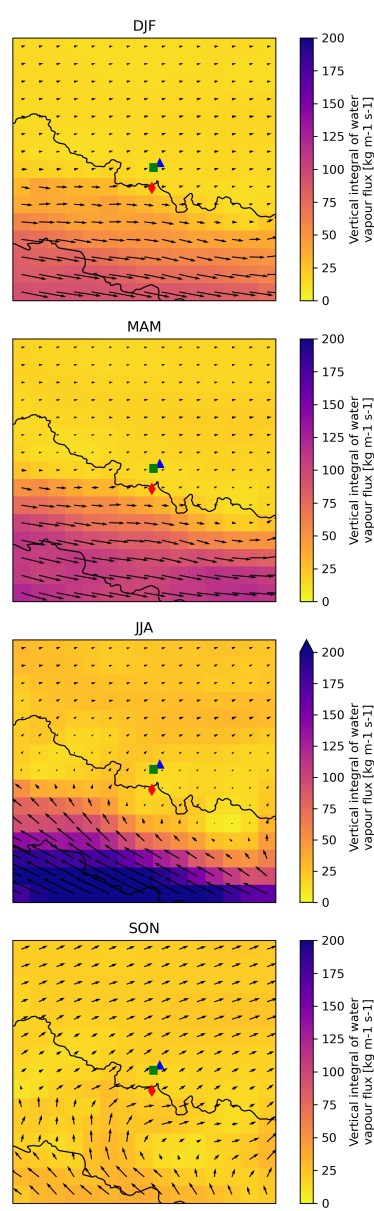

**Figure A1.** Vertical integral of water vapor flux from ERA5 reanalysis for the period 1993-2023. The red diamond shows the location of Yala AWS, the green square Golojang Co and the blue triangle Paiku AWS.



*Author contributions.* Conceptualization: TB, FB, LM and WI; Data Curation: TB (lake pressure data), YL (lake pressure and meteorological
data) and EC (CORDEX-FPS-CPTP data); Formal Analysis: TB; Funding Acquisition: WI and TY; Methodology: HP; Supervision: FB, LM
and WI; Writing: all authors

*Competing interests.* Emily Collier is a member of the editorial board of The Cryosphere.

*Acknowledgements.* Leo Martin is funded by "Chaire de professeur junior" ENVDYN from the French Ministry of Research.



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
