# Peer review of "Brief communication: Sharp precipitation gradient on the southern edge of the Tibetan Plateau during cold season"

_EGUsphere, 2025_

## Author Comment (AC2)

**Answer to Referee 1**

We are very grateful to Reviewer 1 for the in-depth reading and for the relevant review we received. We present below our detailed answer to the discussed points. The reviewers' comments appear in orange and our responses appear in black.

0.

This paper presents in situ observed precipitation at three locations in the central Himalaya and compares these in situ precipitation measurements to two simulated estimates. One of the in situ measurement uses a novel technique where precipitation is derived from pressure measurements taken from a frozen lake. The results demonstrate a large gradient in the observed precipitation across a very small distance, and briefly discuss the ability of the simulated datasets to represent those gradients.

Generally it's well known that large windward/leeside gradients can exist in regions where the precipitation is strongly modified by terrain, and I think this could be better referenced in this paper. However, the novelty here is in the use of the frozen-lake-measured precipitation and in particular its utility in such a data-sparse environment, and the test and figures are generally well presented. I have several specific comments below that would improve the paper that I recommend addressing before acceptance.

We thank the reviewer for their positive appreciation of our work. We are grateful for these scientifically relevant comments and we will attempt here to respond to them as best we can.

1.

Figure 1 comment 1: It would be very helpful for the reader less familiar with the region to include a thumbnail/inset panel showing the broader location of this (fairly small) geographical location

We added a subfigure on figure 1, allowing the reader to locate the study sites:

[Figure]

2.

Figure 1 comment 2: I initially didn't understand why the three in situ markers on the colorbars of b and c shifted values between the panels, then I realized that (I think) these represent the model values at the three closest gridpoints to the in situ locations. This should be made more clear in the Figure caption, and a suggestion is to use outlined markers (diamond, square, and triangle for Yala, Golojang, and Paiku, respectively) rather than filled markers on panels b and c and the colorbars.

We understand that the caption lacks clarity. We modified the caption to:

" Figure 1. (a) Location of study sites. (b) Elevation and (c & d) spatial distribution of total precipitation over the study area from CPTP and ERA5-Land models from 5 december 2019 to 15 July 2020. On the maps, the symbols show the locations of the in-situ recordings. On the color bars on the right side of the maps, symbols indicate interpolated values of precipitation at the study sites. The solid lines mark the boundary of the Langtang National Park and Paiku Co basin."

We tried to plot outline markers instead of filled markers, but it was not successful in improving the readability of the figure as the markers were not visible above the raster layer. We choose to keep the color theme of the study sites that is used in all three figures. We clarified the caption to make it more explicit.

3.

Figure 1 comment 3: I suggest you add the location of Paiku CO (which I believe is slightly off the northern end of the map) to these figures.

We added the lake on the figure (see answer to Figure 1 comment 1).

4.

Comment 1: 118-122 and L. 158-159: The discussion/comparison to Paiku Co values (which are not shown but referenced from Lei et al. 2021) is confusing, because Paiku Co is not shown on the paper's map, and it's unclear why the data from that site are not used in the analysis. If they are available, including the Paiku Co data would be a strong addition to the paper. If not available, this should be stated more explicitly, and any differences between the datasets (e.g., period of overlap, or methodological differences, etc) should be more clearly stated as well.

We agree that the text lacks clarity. We added Paiku Co level change values from the series of Lei et al. (2021). We chose not to include precipitation data collected in villages close to Paiku Co shores because they are measured with unshielded gauges which we find less reliable than the shielded OTT2.

- 118-122: We added the amplitude and the rate from Lei's 2021. modify to: "Golojang Co seasonal level cycle is more pronounced than the cycle of Paiku Co (Lei et al., 2021). During 2013 to 2017, Paiku Co water level had an average amplitude of 51 cm, and had a near constant decrease rate of 7 cm month$^{-1}$ during the post monsoon and the winter on average. Paiku Co level rises sharply at the end of pre-monsoon (Lei et al., 2018), but its level decreases slower than Golojang Co during post-monsoon with a decrease rate greater than 23 cm month$^{-1}$ during 2021 and 2022, and a much lower rate of 4 cm month$^{-1}$ during the winter."

- 158-159 modified to: "While the Yala and Golojang Co snowfall observations are similar, the much lower precipitation totals recorded at Paiku AWS have the same order of magnitude as the 150 mm – 200 mm precipitation previously measured close to Paiku Co (Lei et al., 2018)"

5.

Comment 2: 190-199: The text in this paragraph meanders and is a bit challenging to follow. I suggest dividing it into two paragraphs, one which describes the comparison of in situ to models and one which describes the potential differences between the AWS and PT methods (and the latter's potential to miss smaller snowfall events)

190-199 This paragraph is indeed too complex to be clear. We divided it into three smaller ones. The first two about the comparison of the in-situ and the models, and the third one about the potential source of error of the PT. This part now reads:

180 : *"ERA5-Land has a very good estimation of the cumulative DJFMAM precipitation in Yala since the in situ is, on average, only 8% greater than the AWS during 2020-2021 and 2021-2022 (Fig. 3 - e and f). Unfortunately, Yala AWS was not working during most of the year of the CPTP simulation, and completely missed winter precipitation as it started recording from 27 March 2020. For the period with recording (from 27 March 2020 to 31 May 2020), Yala AWS recorded 241 mm of precipitation (Fig. 3a), against 76 mm estimated by CPTP, which is a large underestimation (Fig. 3g).*

*At Golojang Co, CPTP provides a DJFMAM cumulative snowfall estimate that differs from the PT estimation by only 1.2% during 2019 - 2020, although CPTP slightly overestimates precipitation at the beginning of the study period and underestimates precipitation at the end compared to in situ (Fig.3 - g). ERA5-Land shows much lower precipitation at Golojang Co, with PT estimates being on average 1230% greater for the three years (Fig. 3d, e, and f).*

*Although we use Golojang PT estimates as references here, they are likely underestimated, as the pressure time series primarily captured wet spells averaging 73 hours, while shorter snowfalls may have been masked by noise. For comparison, during the period from 01 December 2020 to 31 May 2021 (the only complete meteorological year during which all sensors were working), there was 26 events recorded at Yala site by the AWS, of which 18*

*lasted less than 2 days and had an accumulation less than 10 mm of snow. At Golojang, the pressure time series allowed us to identify 23 events in the same period, of which only 7 could be matched with Yala small events."*

6.

209-210: While the larger gradients in CPTP may be promising, it's difficult to know whether the even larger values in CPTP to the west and south of Yala AWS are realistic or overdone (while numerical models can accurately depict orographic precipitation, some studies have shown wet biases in high terrain, e.g., He et al. 2019) – perhaps some comment along these lines could be added.

209-210: modified to:

" […] because our results suggest that such gradients can exist. Previous studies demonstrated that high resolution modeling can present large positive biases (He et al. 2019). In situ recordings are still needed to determine whether the large gradients around the study area are realistic or exaggerated, even though observational networks are never perfectly fitted to evaluate model precipitation (Lundquist et al., 2019).

7.

218: These results are not shown and thus should, at a minimum, be stated as such. However, I would encourage adding these results, either as an additional figure, panel in an existing figure, or to Figure A1

This section indeed lacks data. As the brief communication are limited to only 3 figure, it will be added as supplementary figure A2:

[Figure]

Supplementary figure A2: Comparing precipitation records using 1 to 1 ratios of mean total monthly precipitation at different locations. Mean ratio of total precipitation are calculated for each month during 1992 – 2021 in ERA5-Land dataset. The error bars represent ± 1 std of the monthly precipitation during the 30 years period.

8.

222: I suggest citing figure A1 somewhere in this sentence

222 Modify to:

   "[…] the monsoon, then increasing back progressively (fig A1)."

9.

231-232: 'However… magnitudes.': I suggest rewriting this sentence somewhat for clarity, e.g., 'It's possible smaller events were not detected because of their small signal compared to large noise in the measurements.'

231-232: modify to:
   'Smaller events may not have been detected because their signal was too weak compared to the measurement noise.'

10.

237: 'despite it is located much closer to the lake.' – please rewrite for clarity, e.g., 'Golojang is closer to Paiku AWS than to Yala AWS'

We have modified it to:
   "Golojang Co cumulative snowfall during DJFMAM (200–400 mm) is closer to Yala AWS than to Paiku Co (5–35 mm), despite its proximity to Paiku Co."

11.

1. 7: 'gradients marks' -> 'gradient marks'. As written this phrase contains a subject-verb agreement error (gradients is plural, marks is singular) – several errors of this type appear in the paper, and hereafter I simply note them without description

2. 217: 'thirty year' -> 'thirty years'

3. 218: 'precipitation… reveal' -> 'precipitation … reveals'

4. 232: 'signals'->'signal'

We have corrected all these points. Thank you for pointing them out.

References:

Lundquist, J., Abel, M. R., Gutmann, E., & Kapnick, S. (2019). Our Skill in Modeling Mountain Rain and Snow is Bypassing the Skill of Our Observational Networks. Bulletin of the American Meteorological Society, 100(12), 2473-2490. https://doi.org/10.1175/BAMS-D-19-0001.1

---

## Author Comment (AC3)

**Answer to Referee 2**

We are very grateful to Reviewer 2 for the in-depth reading and for the relevant review we received. We present below our detailed answer to the discussed points. The reviewers' comments appear in orange and our responses appear in black.

0.

This manuscript proposes the use of vapor pressure over frozen lake surfaces to estimate cold-season snowfall, offering a practical approach to address the challenges posed by limited and difficult in-situ snowfall observations over lake areas. While the study presents certain merits and has reference value for related hydrometeorological research, several key issues must be addressed before it can be considered for publication.

We thank the reviewer for their positive appreciation of our work. This is likely due to a lack of precision in our manuscript, but we feel that there is a slight misunderstanding in the reviewer interpretation of our work: we measure the *lake water pressure* as a proxy of snow overburden, and not *vapor pressure* over frozen lake. We clarified this point in the text to avoid confusion: l21: "*To overcome these issues, recent studies suggested that solid precipitation could be also estimated from frozen lake* water *pressure changes (Pritchard et al., 2021)*". L69: "*Three Hobo U20* water *pressure transducers (PT) were immersed in Golojang Co*". L75: "I*t has been demonstrated that such* water *pressure time series can be interpreted as direct measurements of the precipitation falling onto the lake during winter-like conditions (Pritchard et al., 2021)*"

1.

The authors repeatedly highlight the use of "new observations." However, both the methodological framework and the ground-based measurements employed are not novel in the field. The authors should clearly articulate what is truly innovative about their observational strategy or data application, and avoid overstating the originality.

As referee 2 stated, both the methodological framework and the type of ground-based measurements employed are indeed not novel in the field. The novelty here lies in the observations themselves, as mentioned on l 25*: "In this study, we report new observations of lake level changes during the cold season"*. These original lake pressure observations are then used to derive snowfall measurements. To avoid confusion, we modified l 227: "*In this communication, we studied the hydro-meteorology of the northern Langtang National Park in Nepal and the Southern Paiku Co basin in Tibet using in-situ and modeled data from conventional and*  *recent methods.*" The two previous mentions of novelty were the only ones present in the manuscript.

2.

The concept of "winter precipitation transition" is central to the manuscript, yet it lacks a clearly defined temporal scale. It remains unclear whether this refers to intra-annual seasonal transitions (e.g., winter to pre-monsoon) or to interannual variability in winter precipitation. This ambiguity persists in the title, abstract, and main text. The authors must explicitly clarify the temporal framework throughout the manuscript. Additionally, the pronounced increase in precipitation from winter to pre-monsoon is a well-recognized climatological feature in many regions,

particularly across the southern Tibetan Plateau. Prior studies have documented a spring precipitation peak in this region. Therefore, the transition described in Golojiang is not a unique phenomenon and should be contextualized within broader regional precipitation dynamics.

We apologize for the lack of precision in our manuscript. Our study tackles only spatial gradients of simultaneous precipitation measurements, and does not investigate temporal variability. The methodological constraints imply that the lake and catchment have to be frozen and hence can only be applied during the coldest months of the year (DJFMAM), but we do not investigate the precipitation increase during pre-monsoon that is indeed a well-documented phenomenon in the region. We implemented the following changes for the sake of clarification:

    - title: " Brief communication: Sharp precipitation gradient on the southern edge of the Tibetan Plateau during cold season"

    - abstract: "We show that precipitation totals can vary by one order of magnitude over a short distance of 10 km in a rather smooth terrain during cold season."

    - text: we replaced "transition" by "gradient" (L161)

3.

The manuscript emphasizes that using lake surface vapor pressure can help identify snowfall events that may be missed by conventional precipitation measurements. However, the authors should also assess whether this method can reliably capture larger-scale precipitation events typically recorded by standard instruments. A more comprehensive evaluation of the method's strengths and limitations is necessary to support the reliability and generalizability of the results.

As mention earlier, we are sorry for the visible misunderstanding due to the lack of clarity of this manuscript. As clarified previously in this response letter, our manuscript does not discuss the use of the lake vapor pressure. However, we use a recent but published method developed by Pritchards et. al 2021 (https://doi.org/10.1175/JHM-D-20-0206.1), allowing to use the water pressure of the lake as an efficient way for recording the precipitation during cold seasons. Here, we apply, as users, this method and discuss its apparent limitations for the present case.

We agree that the magnitude of the events that can be detected by a novel method is an important question. The method seems reliable to detect large snowfall events and overall season total snowfall with a relatively low errors (l135 : "*The total precipitation have been estimated to 420 ± 46 mm for the period 01 December 2019 to 31 May 2020, 307 ± 27 mm for the period 01 December 2020 to 31 May 2021, and 211 ± 9 mm for the period 01 December 2021 to 14 April 2022 (Fig. 3 - a, b and c)*"), but is more questionable for small events (see response to referee #1's 5th comment: "*Although we use Golojang PT estimates as references here, they are likely underestimated, as the pressure time series primarily captured wet spells averaging 73 hours, while shorter snowfalls may have been masked by noise*").

4.

The manuscript presents an interesting approach, but substantial revisions are required to clarify its novelty, conceptual framework, and methodological reliability. I encourage the authors to address the concerns raised above to enhance the scientific clarity and impact of the study.

We are very grateful for the referee's comments and believe that our detailed responses address their concerns.

References:

Pritchard, H. D., D. Farinotti, and S. Colwell, 2021: Measuring Changes in Snowpack SWE Continuously on a Landscape Scale Using Lake Water Pressure. J. Hydrometeor., 22, 795–811, https://doi.org/10.1175/JHM-D-20-0206.1.

---

## Author Response (AR1)

Dear Thomas,

We thank you sincerely for your thorough reading of our answers and for your relevant comments. Please find bellow a point-by-point response to all the comments. Editor comments are in tables with blue background, and our responses are in tables with orange background.

All the best,

Titouan and co-authors.

Comments from the handling editor (Thomas Mölg)

EC1: Modified Figure 1 appears to have two panels labelled (c) - please correct.

**REC1**: The error was corrected

**EC2**: Figure A2 caption: I feel the description of what the "ratio" means is overcomplicated/unclear. Please try to define the ratio in even simpler terms.

**REC2**: The caption indeed lacks of clarity. Modified to:

"Figure A2: Ratios of mean monthly precipitation between different locations (Paiku/Yala, Golojang/Yala, Golojang/Paiku), calculated from ERA5-Land data for the period 1992–2021. Each point represents the mean ratio for a given month, and the error bars indicate ±1 standard deviation of monthly precipitation over the 30-year period"

**EC3**: "even though observational networks ... (Lundquist et al., 2019)." -- I am not sure that what you write is really the key statement of the Lundquist paper. Please re-think this half sentence critically and modify if needed. You probably have to define "observational" more precisely (I guess you mean in-situ measurements?

**REC3**: You are right that our citation of Lundquist et al. 2019 did not fit with the main message of the paper. We rewrote the sentence and removed the reference to the Lundquist paper: I205" [...] because our results suggest that such gradients can exist. Previous studies demonstrated that high resolution modeling can present large positive biases (He et al. 2019). More in situ recordings are still needed to determine whether the large gradients in CPTP product around the study area are realistic or exaggerated, although in situ observation networks are never perfectly fitted to evaluate model precipitation."

In addition, we added another mention to orographic precipitation as suggested by referee #1 in the introduction of they comment:

I200 "In particular, they show relatively high precipitation on the highest parts of the study area, mainly above 5000 m a.s.l., and on the southern slopes of the Himalayas, as expected for orographic precipitation (Roe et al. 2005)."

---

## Author Response (AR2)

**Editor's comments**

We are very grateful to the editor for the in-depth reading of our manuscript and for all the efforts he put in the review of our work. We present below our detailed answer to the discussed points. The editor's comments appear in orange and our responses appear in blue.

EC0: We must address one technical matter, however, before formal acceptance. As noted by the publisher, the length of your manuscript exceeds the "brief communication" extent quite a bit, especially with the addition of new panels/figure in the revision. My advice and requests would be as follows.

AEC0: The length of the manuscript exceeds indeed the specified size. We did our best to shorten it without losing any information or clarity.

EC1: Please go through the text carefully and delete redundant sentences or words. No significant text parts, of course, but every text typically has room for shortening. Ask the native speakers in the author team to shorten some unnecessarily long expressions.

AEC1: Each paragraph has been revised and rewritten more effectively in order to reduce their size to the minimum possible.

EC2: Go through your references. Especially those cited only one time in the text could be candidates for deletion.

AEC2: We believe that none of our references can be removed, as those cited only once are either data sources or references implemented at the suggestion of the referees.

EC3: Remove the appendix and put the two figures in a supplement (and then refer to Figure S1 and S2 in your main text). Note that in TC an appendix (= part of the main manuscript) is not the same as a supplement (= separate file containing supplementary material). Your response to one reviewer sounded to me as if you equate the two.

AEC3: There was indeed confusion. We removed the appendix figures and added them as assets for the manuscript as supplementary figures.

**Editorial team's comment**

We are grateful to the editorial team of The Cryosphere for all their contributions to our work. We present below our detailed answer to the discussed point. The Editorial team's comments appear in orange and our responses appear in blue.

ETC1: Please ensure that the colour schemes used in your maps and charts allow readers with colour vision deficiencies to correctly interpret your findings. Please check your figures using the Coblis – Color Blindness Simulator (https://www.color-blindness.com/coblis-color-blindness-simulator/) and revise the colour schemes accordingly with the next file upload request. -> Fig. 3

AETC1: We apologize if our color scheme is confusing for colorblind people. Since this issue was already pointed out in a previous editorial team comment, we added, on the first correction of the manuscript, small pictograms on the figure 3 (square, dot & triangle) allowing readers to understand the figure without using the color scheme.